# Ultrasound Neuromodulation Reduces Demyelination in a Rat Model of Multiple Sclerosis

**DOI:** 10.3390/ijms231710034

**Published:** 2022-09-02

**Authors:** Feng-Yi Yang, Li-Hsin Huang, Meng-Ting Wu, Zih-Yun Pan

**Affiliations:** 1Department of Biomedical Imaging and Radiological Sciences, National Yang Ming Chiao Tung University, Taipei 112, Taiwan; 2Division of Neurosurgery, Cheng Hsin General Hospital, Taipei 112, Taiwan

**Keywords:** demyelination, remyelination, ultrasound, multiple sclerosis, microglia, oligodendrocyte

## Abstract

Microglia, astrocytes, and oligodendrocyte progenitor cells (OPCs) may serve as targets for remyelination-enhancing therapy. Low-intensity pulsed ultrasound (LIPUS) has been demonstrated to ameliorate myelin loss and inhibit neuroinflammation in animal models of brain disorders; however, the underlying mechanisms through which LIPUS stimulates remyelination and glial activation are not well-understood. This study explored the impacts of LIPUS on remyelination and resident cells following lysolecithin (LPC)-induced local demyelination in the hippocampus. Demyelination was induced by the micro-injection of 1.5 μL of 1% LPC into the rat hippocampus, and the treatment groups received daily LIPUS stimulation for 5 days. The therapeutic effects of LIPUS on LPC-induced demyelination were assessed through immunohistochemistry staining. The staining was performed to evaluate remyelination and Iba-1 staining as a microglia marker. Our data revealed that LIPUS significantly increased myelin basic protein (MBP) expression. Moreover, the IHC results showed that LIPUS significantly inhibited glial cell activation, enhanced mature oligodendrocyte density, and promoted brain-derived neurotrophic factor (BDNF) expression at the lesion site. In addition, a heterologous population of microglia with various morphologies can be found in the demyelination lesion after LIPUS treatment. These data show that LIPUS stimulation may serve as a potential treatment for accelerating remyelination through the attenuation of glial activation and the enhancement of mature oligodendrocyte density and BDNF production.

## 1. Introduction

Multiple sclerosis (MS) is the most common demyelinating disease of the central nervous system (CNS) and is characterized by oligodendrocyte death, myelin loss, and disrupted action potential conduction [1]. MS can be classified according to its disease course into relapsing-remitting MS (RRMS), primary progressive MS (PPMS), and secondary progressive MS (SPMS) [2]. The pathogenesis of MS is characterized by a series of pathobiological events, ranging from microglia activation to demyelination and axonal degeneration [3,4]. Neurodegeneration in MS is commonly considered to be associated with the development of an inflammatory autoimmune response in the CNS [5]. Inflammation not only results in demyelination but also induces neuronal damage [6,7]. Although the underlying mechanism of MS remains unclear, the existing drugs used to treat MS target inflammation [8,9]. However, current efforts are focused on developing treatment approaches that utilize several combinations of compounds with different mechanisms of action with the aim of promoting remyelination [10]. Currently, more than a dozen drugs are approved for RRMS and one agent for PPMS [11]. Ocrelizumab, a monoclonal antibody, was the first medication to be approved for PPMS [12].

Remyelination in the CNS is a complex process that requires the coordination of multiple resident cell types, including oligodendrocyte progenitor cells (OPCs), neuronal axons, microglia, and astrocytes, all of which may be targets for enhanced remyelination [13]. Targeting these cells, along with the currently available disease-modifying therapies, has the potential to develop a therapeutic strategy to treat progressive forms of MS, which currently lacks effective therapies. Furthermore, the brain-derived neurotrophic factor (BDNF) levels are decreased in patients with MS [14]. Astrocyte-derived BDNF may represent a source of trophic support that can reverse the deficits following demyelination [15]. Enhancing the endogenous sources of trophic support is crucial in the CNS, because the administration of exogenous proteins to the brain is challenging [16,17].

Our previous studies reported that low-intensity pulsed ultrasound (LIPUS) was able to alleviate neuroinflammation by inhibiting Toll-like receptor 4 (TLR4)/nuclear factor kappa B (NF-κB) signaling and enhancing cAMP response element-binding protein (CREB)/BDNF expression in lipopolysaccharide-treated mice [18]. LIPUS stimulation upregulates BDNF production in astrocytes through the activation of NF-κB via the tropomyosin receptor kinase B (TrkB)–phosphoinositide 3-kinase (PI3K)–protein kinase B (Akt) and calcium–Ca^2^/calmodulin-dependent protein kinase (CaMK) signaling pathways [19]. The anti-inflammatory effects and neuroprotective modulations associated with LIPUS applications in in vitro and in vivo studies indicate that transcranial ultrasound stimulation (TUS) might have potential therapeutic effects in patients with brain disorders [20,21,22,23]. Additionally, TUS treatment resulted in a clear but variable degree of myelin recovery when used to treat vascular dementia and AlCl_3_-treated rats [24,25]. Another study has demonstrated that TUS is capable of accelerating remyelination of MS lesions [26]. However, the underlying mechanisms of TUS-enhanced remyelination are still unknown.

The purpose of this project was to study the effects of TUS on remyelination and the inflammatory response associated with demyelination, which involves the activation of astrocytes and microglia [27,28]. LPC causes toxicity in myelin and is used to induce local demyelination in vivo [29,30,31]. TUS was exploited as a form of physical therapy to study the effects of ultrasound neuromodulation during remyelination. We histologically confirmed the occurrence of remyelination using immunohistochemistry (IHC) staining. Our results suggest that TUS enhances remyelination in the hippocampus of rats by upregulating the BDNF levels and inhibiting neuroinflammatory activity.

## 2. Results

### 2.1. Ultrasound Treatment Enhances Remyelination

Remyelination in the demyelination lesion was monitored using the axonal marker NF (green) and the myelin sheath marker MBP (red) 7 days post-injection. The LIPUS treatment group showed better axonal remyelination compared with the sham group (Figure 1A). The quantification of NF^+^ axons and MBP^+^ staining showed that the percentages of axons and myelin sheaths significantly decreased in the LPC-only group compared with both the sham and LPC+LIPUS groups (Figure 1B,C). The evaluation of juxtaposed NF^+^ axons and MBP^+^ staining showed that the percentage of axons surrounded by myelin sheaths significantly increased in LPC+LIPUS rats compared with LPC-only rats (0.83 ± 0.08 versus 0.13 ± 0.02, *p* < 0.001; Figure 1D). Similarly, the remyelination index was significantly higher in LPC+LIPUS rats compared with LPC-only rats (0.12 ± 0.01 versus 0.02 ± 0.00, *p* < 0.001; Figure 1E).

### 2.2. Ultrasound Treatment Reduces Astrocytic Activation and the Density of Microglia in the Vicinity of the Demyelination Lesion

Demyelination is accompanied by neuroinflammation, which involves the activation of astrocytes and microglia, as shown in Figure 2A,B, respectively. LPC-induced demyelination resulted in a significant increase in astrocytic cell density (*p* < 0.05, Figure 2C), which was significantly decreased after LIPUS treatment (*p* < 0.01, Figure 2C). Furthermore, the fraction of the total area covered by GFAP^+^ cells (*p* < 0.01, Figure 2D) and the optical density (OD) of GFAP^+^ staining (*p* < 0.05, Figure 2E) significantly increased in the hippocampus of LPC-treated rats. The fraction of the total area covered by GFAP^+^ cells (*p* < 0.01, Figure 2D) and the OD of GFAP^+^ staining (*p* < 0.05, Figure 2E) decreased significantly following the LIPUS treatment compared with the LPC-only treatment. In the sham rats, saline injection into the hippocampus showed mild microglial activation (Figure 2F). However, a significant increase of activated microglia was observed in the LPC-injected hippocampus compared with the saline-injected hippocampus (1052.85 ± 208.07 versus 67.79 ± 8.39, *p* < 0.001; Figure 2F). Microglial activation was manifested as an increase in the microglial density (Iba1^+^ cells). LIPUS treatment significantly decreased the microglial density in the demyelination lesion compared with the LPC treatment alone (213.45 ± 54.46 versus 1052.85 ± 208.07, *p* < 0.05; Figure 2F).

### 2.3. Microglial Phenotypes in the Demyelination Lesion of the Hippocampus

Iba1 staining was used to identify and quantify the distribution of microglial cells, and microglial morphology was used to classify microglia into ramified, hypertrophic, dystrophic, rod-shaped, and amoeboid microglia, as previously described (Figure 3A) [32]. In the demyelination lesion of the hippocampus, the number of microglia was counted in five randomly placed ROIs. The quantification of Iba1^+^ cells in the LPC group demonstrated a significant decrease (*p* < 0.001, Figure 3B) in the number of ramified cells, whereas the number of hypertrophic, dystrophic, rod-shaped, and amoeboid microglia significantly increased (all *p* < 0.05, Figure 3C–F) compared to both the sham and LPC+LIPUS groups. The numbers of each of the five microglial subtypes were plotted as a percentage of the total microglial number to visualize the distribution of microglial morphologies among the different groups (Figure 3G).

### 2.4. Ultrasound Treatment Enhances the Maturation of Oligodendrocytes

OPCs are capable of proliferating and differentiating into mature oligodendrocytes, which then produce the myelin sheaths that insulate axons. Here, we explored the influence of LIPUS on oligodendrocyte maturation. LPC injection resulted in a significant increase in the cell density of Olig2^+^ compared with a saline injection (275.61 ± 44.74 versus 77.38 ± 6.73, *p* < 0.001; Figure 4A,B). LIPUS treatment significantly reduced the density of Olig2^+^ cells in demyelination lesions compared with the LPC-only treatment (100.03 ± 9.90 versus 275.61 ± 44.74, *p* < 0.001; Figure 4A,B). By contrast, LPC injection led to a significant reduction in the cell density of CC-1^+^ cells compared with the saline injection (199.89 ± 31.90 versus 280.57 ± 24.48, *p* < 0.001; Figure 4A,C). LIPUS treatment significantly increased the density of CC-1^+^ cells in the demyelination lesion compared with the LPC-only treatment (376.96 ± 10.68 versus 199.89 ± 31.90, *p* < 0.001; Figure 4A,C). Statistical analysis revealed that the LPC treatment induced a decrease in BDNF expression in the demyelination lesion compared with the saline treatment (0.30 ± 0.16 versus 1.00 ± 0.20, *p* < 0.01; Figure 4D,E). Compared with the LPC-only group, a significant increase in BDNF expression was observed in the LPC+LIPUS group (1.43 ± 0.43 versus 0.30 ± 0.16, *p* < 0.001; Figure 4D,E).

## 3. Discussion

We found that LIPUS stimulation enhanced remyelination following the induction of a local demyelination lesion in the hippocampus of LPC-treated rats. The current study results showed that LPC induction followed by LIPUS treatment enhanced the extent of remyelination in the hippocampus and enhanced the expression of MBP and CC-1 compared with LPC treatment alone. We confirmed that LIPUS treatment reduced the activation of astrocytes and microglia and promoted BDNF expression (Figure 5).

Substantial evidence suggests that glial activation and neuroinflammation play critical roles in the pathogenesis of MS [6]. Myelin damage is associated with the activation of astrocytes and microglia, which are known to be involved in the pathogenesis of MS. Microglial cells were characterized in this study by manually assessing the morphology of microglial cells following immunostaining for Iba1 (Figure 3A). Figure 3G shows that the LPC group was characterized by a significant decrease in the population of ramified cells in the demyelination lesion compared with the sham group. The dystrophic and amoeboid phenotypes, which are typically associated with inflammatory responses [33,34], were significantly more abundant in the LPC group than in the sham group. The LPC+ LIPUS group exhibited an increase in ramified microglia and fewer amoeboid cells than the LPC group. Compared to the LPC group, the hypertrophic and rod-shape phenotypes were significantly decreased in the LPC+LIPUS group. These findings indicate that a heterologous population of microglia with various morphologies can be found in the demyelination lesion after LIPUS treatment, which may potentially promote neuroprotective effects in MS.

Olig2 plays stage-specific roles, mediating opposing functions during the differentiation and maturation of oligodendrocytes [35]. During the early stages of OPC differentiation, Olig2 may act as an activator; however, the deletion of Olig2 during later stages of oligodendroglial development may represent an effective strategy for promoting remyelination. In recent years, transcranial magnetic stimulation, a type of noninvasive brain stimulation, has shown as a potential therapy for demyelination disorder by enhancing the differentiation of OPCs [36]. Moreover, transcranial direct current stimulation could mobilize OPCs towards the ischemic lesion [37]. Consistent with this hypothesis, our data revealed that LIPUS treatment enhanced remyelination by accelerating oligodendrocyte maturation (CC-1) rather than by enhancing OPC (Olig2) proliferation 1 week after LPC administration (Figure 4B,C). Compared to a previous study [26], we demonstrated that LIPUS not only increase the myelin in the demyelination lesion but also enhance oligodendrocyte maturation. The previous work was carried out in the cuprizone model and with a different ultrasound protocol. Consequently, it might be that, at different parameters and animal models, different mechanisms exist for coupling acoustic wave into brain activity.

BDNF increases myelinogenesis and axonal regeneration by promoting the differentiation of OPCs [38]. The upregulation of MBP expression is thought to be activated by BDNF signaling to enhance the differentiation of OPCs. A significant increase in oligodendrocyte proliferation is observed following BDNF administration in cases of spinal cord injury [39]. We observed an increase in BDNF (Figure 4E) and MBP (Figure 1C) expression following the LIPUS treatment. A bunch of evidence indicates that BDNF can promote myelin regeneration in different animal models of demyelination [40,41]. Based on previous studies in the LPC-induced model, demyelination is the most active process within the first week, and day 7 is considered adequate for evaluating the effect of treatment on demyelination [29,42]. Following LPC administration, the expression of Olig2 was increased on day 7, which showed the recruitment of OPCs into the lesion (Figure 4B). When comparing the expression of Olig2 in the LIPUS-treated group, the level of Olig2 was lower than the LPC group on day 7. On the other hand, the expression of MBP was higher in the LIPUS-treated group on day 7. Therefore, LIPUS may cause an increase in the BDNF level and provide a neuroprotective effect and trophic support for reducing the entity of demyelination.

Although significant advances in the development of remyelinating drugs have been made recently, further work remains necessary to minimize side effects and alleviate safety concerns. In contrast, TUS is a safe and noninvasive form of neuromodulation, which is broadly applied in novel therapeutic methods for initial human studies [43,44,45,46,47,48]. Although increasingly being investigated, TUS is still in its early phases. The ultrasound stimulus is defined by several parameters: operating frequency, intensity, duration, and pulse repetition frequency. Each of these parameters may have a different effect on the experimental outcome. In this study, one set of parameters was selected based on our previous works. Further investigations into the optimal ultrasound parameters are needed. In recent years, it has been shown that TUS exerts a significant therapeutic effect on Alzheimer’s disease, depression, and traumatic brain injuries [49,50,51]. Compared to other noninvasive brain stimulation tools, TUS has deeper penetration and a higher spatial resolution. One limitation of this study was that the use of a single-element transducer made it difficult to provide targeted sonication. A phased-array transducer could be employed to achieve more localized sonication in future works. Additionally, the goal of this study was to investigate the possible effects in the demyelination lesion after LIPUS treatment by histological analysis. Behavioral or motor function will be investigated in the next step.

## 4. Materials and Methods

### 4.1. Toxic Model of Demyelination and Remyelination

Fifteen male Sprague–Dawley rats weighing 280–320 g (age: 12 weeks) were purchased from LASCO. This study protocol was approved by the Animal Care and Use Committee of National Yang Ming Chiao Tung University (No. 1100224). Animals underwent stereotaxic surgery to induce a demyelination by injection of LPC, as previously described [52,53]. A single dose consisting of 1.5 µL of 1% LPC (Sigma-Aldrich, St. Louis, MO, USA) dissolved in 0.9% saline was injected unilaterally into the CA1 region of the right hippocampus at a rate of 0.5 µL/min using the following coordinates: anteroposterior, −3.6 from bregma; mediolateral, +2.6; dorsoventral, −3.2 mm from the dura surface (Figure 6A). In contrast to LPC rats, sham rats received an equal volume of saline solution at the same site.

### 4.2. TUS Setup and Treatment Protocols

The TUS setup was similar to that used in our previous study. LIPUS was generated using a therapeutic ultrasound generator (ME740, Mettler Electronics, Anaheim, CA, USA) and a 1-MHz plane transducer (ME7413: 4.4 cm^2^ effective radiating area; Mettler Electronics, Anaheim, CA, USA) with 2-ms burst lengths at a 20% duty cycle and a repetition frequency of 100 Hz. The spatial average intensity (SAI) over the plane transducer head was 500 mW/cm^2^. The transducer was mounted on a removable aluminum cone with a 10-mm diameter at the cone tip. Sonication was precisely targeted using a stereotaxic apparatus (Stoelting, Wood Dale, IL, USA). The acoustic wave was delivered to the demyelination lesion in the brain. Each rat’s right hemisphere was treated with LIPUS using triple sonication. The duration of each sonication was 5 min, with a 5-min interval between each sonication treatment. The optimal parameters of the LIPUS exposures were selected based on the results of our previous studies [24]. Histological evaluations were performed on day seven after LPC administration (Figure 6B).

### 4.3. Tissue Processing

Animals were deeply anesthetized and perfused intracardially with phosphate-buffered saline (PBS), followed by 4% paraformaldehyde. Brains were collected, post-fixed in 4% paraformaldehyde overnight, and transferred to PBS containing 30% sucrose for cryoprotection. Serial coronal sections (10 μm) were obtained using a microtome. The images of three brain sections from the center of the demyelination lesion were acquired in 4 to 5 rats per group for image analysis.

### 4.4. Immunofluorescence

Brain tissues were processed for the immunofluorescent detection of MBP (1:400; Abcam, Cambridge, MA, USA); neurofilament (NF; 1:50; Abcam, Cambridge, MA, USA); BDNF (1:200; Abcam, Cambridge, MA, USA); and markers of microglia (ionized calcium-binding adaptor molecule 1 (Iba1); 1:200; GeneTex, Inc., Irvine, CA, USA), astrocytes (glial fibrillary acidic protein (GFAP); 1:200; GeneTex, Inc., Irvine, CA, USA), OPCs (Olig2; 1:100; Abcam, Cambridge, MA, USA), and mature oligodendrocytes (CC-1; 1:200; Sigma-Aldrich, St. Louis, MO, USA) [27,54]. The brain sections were incubated with primary antibodies overnight, then washed and incubated with either Alexa Fluro 488- or Alexa Fluro 594-tagged secondary antibodies (1:500; Abcam, Cambridge, MA, USA) at room temperature. For the evaluation of NF^+^ and MBP^+^ fibers, six different fields from both edges of the lesion were analyzed in three different sections for each animal. The selected fields were used to measure the fraction of the total area covered by NF^+^ fibers, the fraction of the total area covered by MBP^+^ fibers, and the fraction of the total area covered by juxtaposed NF^+^ and MBP^+^ fibers. The juxtaposition of NF^+^ and MBP^+^ fibers was determined using ImageJ software. The remyelination index was calculated by dividing the value of the juxtaposed NF^+^ and MBP^+^ fraction by the value of the NF^+^ fraction.

Three different sections per rat were analyzed at the same exposure level. At least three randomly distributed fields from the hippocampus were captured for each section. The percentage of the total area occupied by GFAP was used as an index of astrocytic activation. Moreover, astrocytic activation was assessed by measuring the optical density of GFAP immunoactivity, as previously described [55,56]. The number of cells positive for GFAP, Iba1, Olig2, and CC-1 was counted in an area of 1250 × 940 μm^2^ in three nonoverlapping fields in the hippocampus using a magnification of ×200. The mean signal intensities for BDNF and myelin were quantified with Image-Pro Plus software (Media Cybernetics, Silver Spring, MD, USA) in a blinded manner.

### 4.5. Statistical Analysis

All data are presented as the mean ± standard error of the mean (SEM). One-way ANOVA with Tukey’s post hoc test was used to evaluate the histological data. Significance was established at a *p*-value ≤ 0.05.

## 5. Conclusions

This study suggests that LIPUS stimulation can enhance the remyelinating effects by inhibiting glial activation, promoting the release of BDNF and oligodendrocyte proliferation, and ultimately increasing the MBP levels in a LPC-induced model of demyelination. LIPUS treatment enhanced the myelination in specific brain regions, indicating a potential novel therapeutic approach for treating myelin loss in MS. Understanding the changes in microglial morphologies following LIPUS treatment may provide additional insights and new directions for the use and design of LIPUS-based therapies. This study provides a rationale for evaluating the potential clinical application of TUS for the treatment of demyelinating diseases.

## Figures and Tables

**Figure 1 ijms-23-10034-f001:**
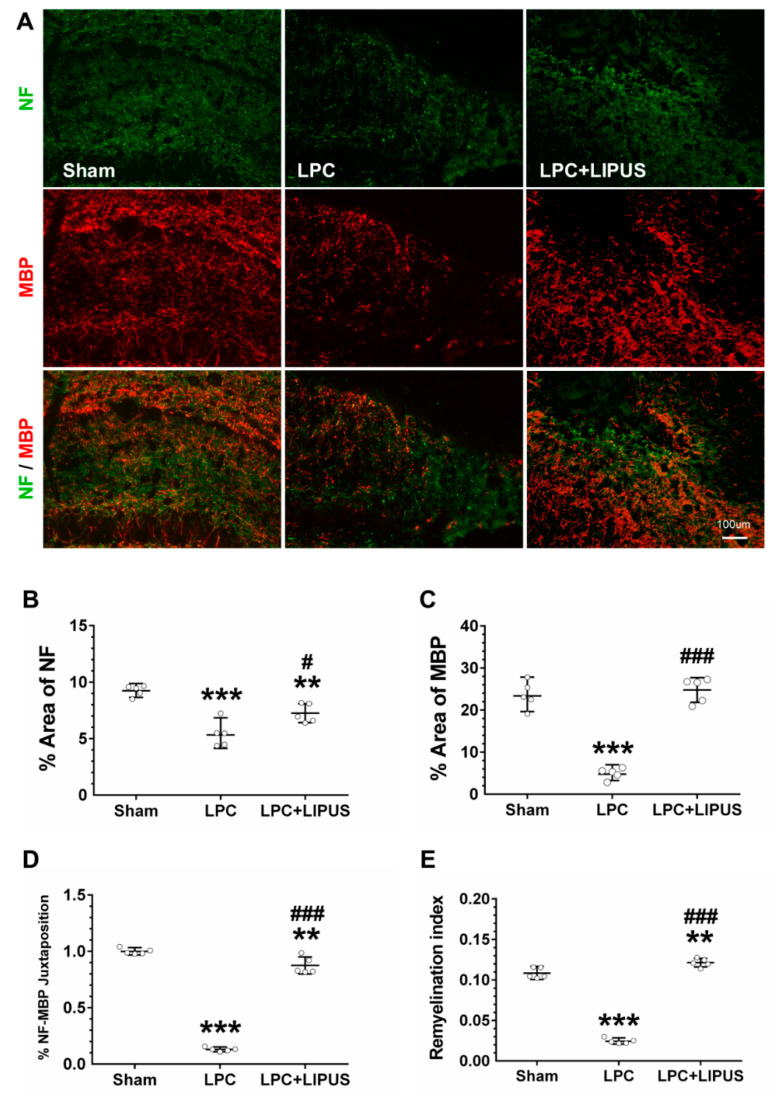
Evaluation of lesion size and axonal integrity 7 days after LPC-induced demyelination. (**A**) Immunofluorescent images of NF (green) and MBP (red) at the demyelination lesion in the hippocampus. (**B**) The percentage area covered by NF^+^ fibers. (**C**) The percentage area covered by MBP^+^ fibers. (**D**) The percentage of juxtaposed NF^+^ and MBP^+^ fibers. (**E**) The LIPUS-treated rats had a significantly higher remyelination index compared with the sham and LPC-treated rats. * and ^#^ denote significant differences from the sham and the LPC groups, respectively (**, *p* < 0.01; ***, *p* < 0.001; ^#^, *p* < 0.05; ^###^, *p* < 0.001, *n* = 5). Scale bar = 100 μm. LPC: lysolecithin, LIPUS: low-intensity pulsed ultrasound, NF: neurofilament, and MBP: myelin basic protein.

**Figure 2 ijms-23-10034-f002:**
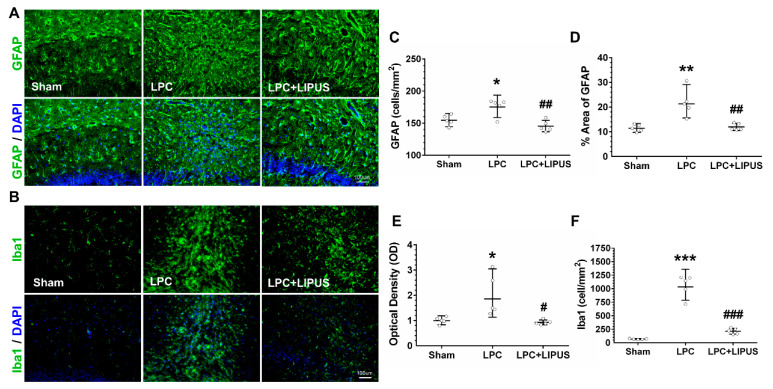
LIPUS treatment reduced astrocytic activation and microglial cell density in the vicinity of the demyelination lesion. Immunofluorescent staining image of the astrocytic marker GFAP (**A**) and the microglial marker Iba1 (**B**). (**C**) LIPUS decreased the density of the astrocytic marker GFAP. LIPUS significantly decreased the percentage area covered by GFAP (**D**) and reduced the GFAP optical density (**E**) in the vicinity of the demyelinated area of the hippocampus in LPC-treated rats. (**F**) The LIPUS group showed a significantly reduced density of Iba1^+^ cells in the vicinity of the demyelination lesion compared to the LPC group. * and ^#^ denote significant differences from the sham and LPC groups, respectively (*, *p* <0.05; **, *p* < 0.01; ***, *p* < 0.001; ^#^, *p* < 0.05; ^##^, *p* < 0.01; ^###^, *p* < 0.001, *n* = 5). Scale bar = 100 μm. LPC: lysolecithin, LIPUS: low-intensity pulsed ultrasound, GFAP, glial fibrillary acidic protein, and Iba1: ionized calcium-binding adaptor molecule 1.

**Figure 3 ijms-23-10034-f003:**
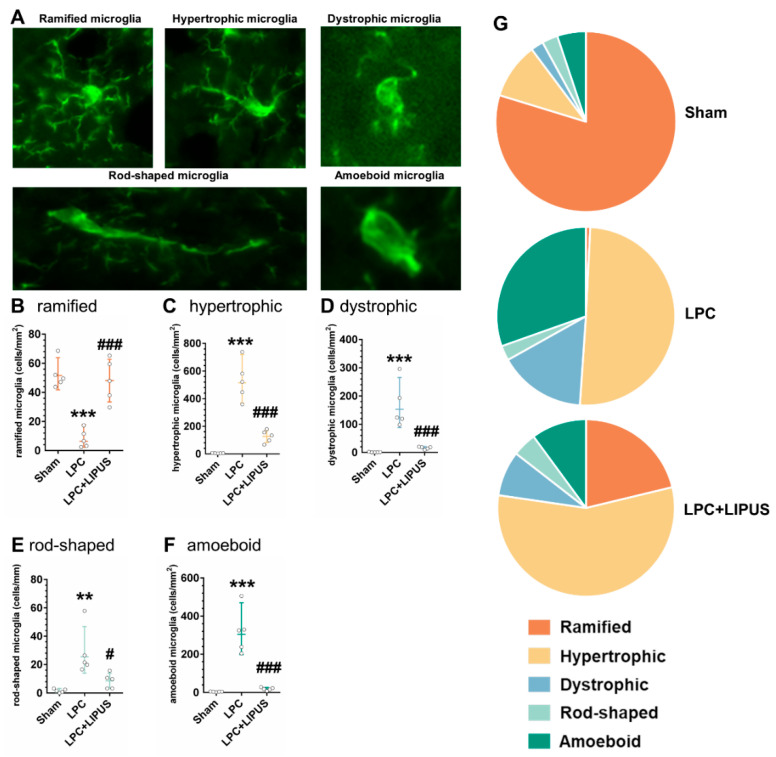
Group-specific patterns in Iba1^+^ microglial morphologies. (**A**) Five microglial morphologies were analyzed: ramified microglia, hypertrophic microglia, dystrophic microglia, rod-shaped microglia, and amoeboid microglia. Microglial counts are provided for the demyelination lesion in the hippocampus for (**B**) ramified, (**C**) hypertrophic, (**D**) dystrophic, (**E**) rod-shaped, and (**F**) amoeboid morphologies. (**G**) The number of microglia in each of the five distinct classes was plotted as a percentage of the total microglial number for the three groups. The magnification is ×200. ^#^ denotes significant differences from the sham and LPC groups, respectively (**, *p* < 0.01; ***, *p* < 0.001; ^#^, *p* < 0.05; ^###^, *p* < 0.001, *n* = 5). LPC: lysolecithin, LIPUS: low-intensity pulsed ultrasound, and Iba1: ionized calcium-binding adaptor molecule 1.

**Figure 4 ijms-23-10034-f004:**
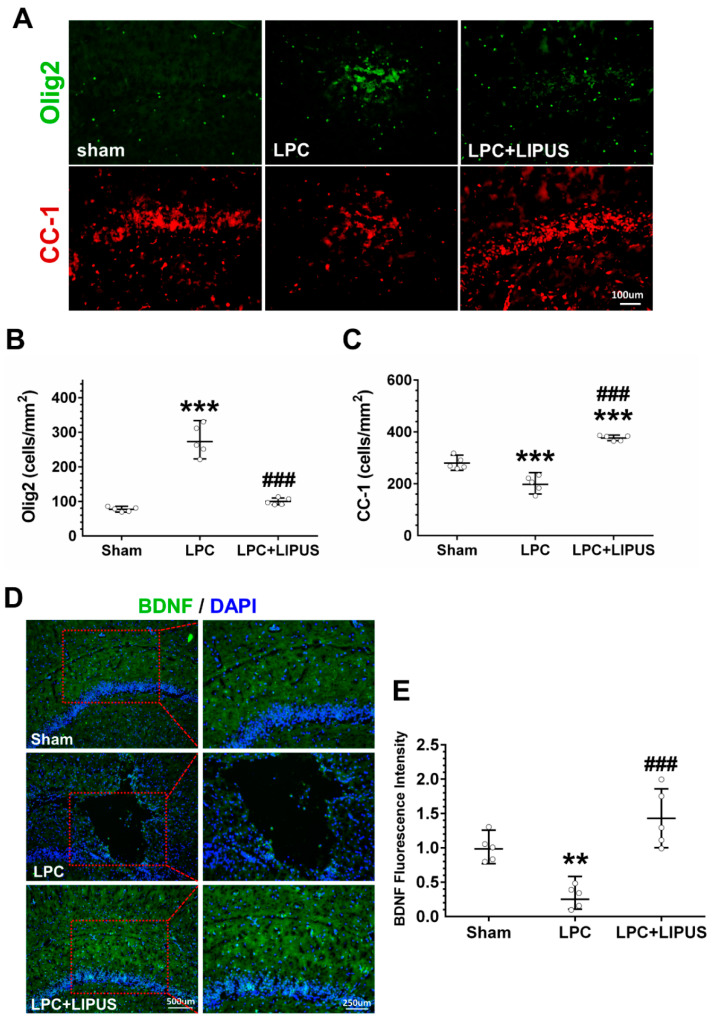
A reduction in Olig2^+^ cells and an increase in CC-1^+^ cells and the BDNF level were observed in LPC-induced lesions following LIPUS treatment. (**A**) Representative immunofluorescent staining of Olig2^+^ (green) and CC-1^+^ (red) in the vicinity of the demyelination lesion. Scale bar = 100 μm. Quantification of (**B**) Olig2^+^ and (**C**) CC-1^+^ cell numbers in the LPC and LPC+LIPUS groups compared to the sham group. (**D**) Representative immunostaining for BDNF in LPC-induced lesions. Left column: scale bar = 500 μm and, in the right column, 250 μm. (**E**) The fluorescence intensities of BDNF were significantly decreased in the LPC group compared with the sham group. Compared with the levels observed in the LPC group, the BDNF expression significantly increased in the LPC+LIPUS group. * and ^#^ denote significant differences from the sham and LPC groups, respectively (**, *p* < 0.01; ***, *p* < 0.001; ^###^, *p* < 0.001, *n* = 5). Scale bar = 100 μm. LPC: lysolecithin, LIPUS: low-intensity pulsed ultrasound, and BDNF: brain-derived neurotrophic factor.

**Figure 5 ijms-23-10034-f005:**
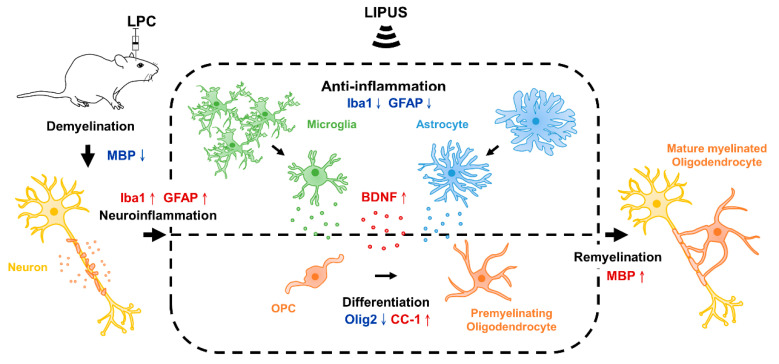
The schematic diagram indicates the potential mechanism through which LIPUS treatment enhances remyelination in a LPC-induced model of demyelination in the hippocampus. Several resident cells impact remyelination after LIPUS treatment, including microglia, astrocytes, oligodendrocyte progenitor cells (OPCs), and oligodendrocytes. Our results demonstrate that LIPUS treatment promotes remyelination in the demyelination lesion. These effects are related to increased expression levels of BDNF and MBP and the inhibition of glial cell activation in LPC-induced demyelination. ↑ and ↓ denote increased and decreased expression, respectively. LPC: lysolecithin, LIPUS: low-intensity pulsed ultrasound, BDNF: brain-derived neurotrophic factor, and MBP: myelin basic protein.

**Figure 6 ijms-23-10034-f006:**
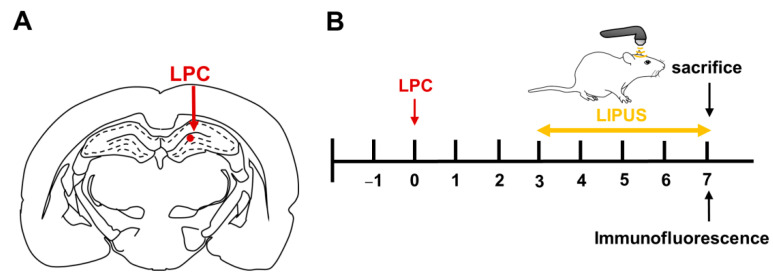
Experimental protocols. (**A**) Schematic diagram of LPC injections into the hippocampus. (**B**) The time course of the study. Rats were treated daily with LIPUS for 5 days and then sacrificed at seven days following LPC administration.

## Data Availability

The data that support the findings of this study are available within the article and from the corresponding author upon reasonable request.

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
