# Peer review of "Ultrasound Neuromodulation Reduces Demyelination in a Rat Model of Multiple Sclerosis"

_ijms, 2022, doi:10.3390/ijms231710034_

Round 1

Reviewer 1 Report

In the paper entitled “Ultrasound neuromodulation enhances remyelination in a rat model of multiple sclerosis”, Yang et al described the effect of LIPUS stimulation on lysolecithin induced demyelination by immunofluorescent analysis on different types of glial cells. The quality of the manuscript is generally good. Data are well presented and interesting. My main concerns are related to the timing of the treatment and how the authors interpretated the results related to OL and myelin.

Main comments:

-Lysolecithin-induced demyelination is a widely used and well characterized model of toxic demyelination. In this model OPC are recruited at the lesion site between 3 and 7 dpl, then they start differentiation and the first sign of remyelination can be observed at 10 dpl. Partial remyelination can be observed at 14 dpl, whereas remyelination is completed at 28 dpl. In this study, LIPUS treatment was performed between 3 and 7 dpl; all the analysis has been performed at 7 dpl. So, based on the aforementioned timing, I believe that the effect of LIPUS on OL population and myelin cannot be interpretated as a pro-remyelinating activity, but rather as an indirect protective activity mediated by the effect of LIPUS on microglia and astrocyte (neuroinflammation). In other words, it seems that LIPUS reduced the entity of demyelination, instead of improved the remyelination. The authors should change the discussion according to these considerations or alternatively produce a new set of data. To assess effects on remyelination the treatment should start at 7 dpl, analyzing data at 10-14 dpl.

-A complete characterization of microglia phenotype after LIPUS treatment is needed. It has been shown that in this model M1 phenotype is necessary for OPC maturation, whereas M2 phenotype is needed for OPC differentiation. A shift from M1 to M2 phenotype induced by LIPUS may explain several of the results (e.g., the reduction of Olig2+ cells coupled to increase in CC1 cells)

Minor comments:

-The authors did not report in which hippocampal regions the analysis have been performed.

-The quality of the BDNF staining is poor. Addition of representative pictures at lower magnification could be useful to evaluate the specificity of the signal.

-In figure 3A the names of the conditions are missing.

Author Response

Comments and Suggestions for Authors

In the paper entitled “Ultrasound neuromodulation enhances remyelination in a rat model of multiple sclerosis”, Yang et al described the effect of LIPUS stimulation on lysolecithin induced demyelination by immunofluorescent analysis on different types of glial cells. The quality of the manuscript is generally good. Data are well presented and interesting. My main concerns are related to the timing of the treatment and how the authors interpretated the results related to OL and myelin.

Explanation:

The manuscript has been revised as suggested. We sincerely appreciate the reviewers’ opinion to largely strengthen our manuscript. The made text changes with bold words are underlined in the amendment part. Moreover, all changes with red words are in the revised manuscript.

Main comments:

-Lysolecithin-induced demyelination is a widely used and well characterized model of toxic demyelination. In this model OPC are recruited at the lesion site between 3 and 7 dpl, then they start differentiation and the first sign of remyelination can be observed at 10 dpl. Partial remyelination can be observed at 14 dpl, whereas remyelination is completed at 28 dpl. In this study, LIPUS treatment was performed between 3 and 7 dpl; all the analysis has been performed at 7 dpl. So, based on the aforementioned timing, I believe that the effect of LIPUS on OL population and myelin cannot be interpretated as a pro-remyelinating activity, but rather as an indirect protective activity mediated by the effect of LIPUS on microglia and astrocyte (neuroinflammation). In other words, it seems that LIPUS reduced the entity of demyelination, instead of improved the remyelination. The authors should change the discussion according to these considerations or alternatively produce a new set of data. To assess effects on remyelination the treatment should start at 7 dpl, analyzing data at 10-14 dpl.

Amendment:

The discussion has been revised as suggested.

“Base on previous studies in LPC-induced model, demyelination is the most active process within the first week and day 7 is considered adequate for evaluating the effect of treatment on demyelination [29, 43]. Following LPC administration, the expression of Olig2 was increased on day 7 which show the recruitment of OPCs into the lesion (Figure 5B). When comparing the expression of Olig2, in LIPUS-treated group the level of Olig2 was lower than LPC group on day 7. On the other hand the expression of MBP was higher in LIPUS-treated group on day7. Therefore, LIPUS may by an increase in BDNF level provide a neuroprotective effect and trophic support for reducing the entity of demyelination.”

-A complete characterization of microglia phenotype after LIPUS treatment is needed. It has been shown that in this model M1 phenotype is necessary for OPC maturation, whereas M2 phenotype is needed for OPC differentiation. A shift from M1 to M2 phenotype induced by LIPUS may explain several of the results (e.g., the reduction of Olig2+ cells coupled to increase in CC1 cells)

Explanation:

Yes. It is an interesting to investigate if a shift from M1 to M2 phenotype can be induced by LIPUS.

Distinct microglia phenotypes predominate at specific stages of focal remyelination. An initial pro-inflammatory microglial phenotype (iNOS, TNF, and CD16-32) is abundant during the early phases of repair when OPCs are recruited to lesions [1]. A subsequent regenerative phenotype (ARG1, IGF1, and CD206) emerges at the onset of oligodendrocyte differentiation and remyelination. In this study, LIPUS treatment was performed between 3 and 7 dpl; all the analysis has been performed at 7 dpl. To test whether LIPUS shift M1 to M2 phenotype a different experimental protocol should be used. This is a good topic and we will design new experimental protocols to investigate it in the future.

[1] Miron, V. E. et al. M2 microglia and macrophages drive oligodendrocyte differentiation during CNS remyelination. Nat. Neurosci. 16, 1211–1218 (2013).

Minor comments:

-The authors did not report in which hippocampal regions the analysis have been performed.

Amendment:

The region of hippocampus has been indicated as suggested.

“…was injected unilaterally into the CA1 region of right hippocampus…”

-The quality of the BDNF staining is poor. Addition of representative pictures at lower magnification could be useful to evaluate the specificity of the signal.

Amendment:

The quality of the BDNF staining has been improved. The representative pictures at lower magnification have been added as suggested.

Figure 5D”

-In figure 3A the names of the conditions are missing.

Amendment:

The names of the conditions in figure 3A have been added as suggested.

“ Sham, LPC, LPC+LIPUS ”

Reviewer 2 Report

The work by Feng-Yi Yang and collaborators investigated the potential use of transcranial ultrasound stimulation (TUS) for multiple sclerosis. To this end, they used rats and induced demyelination in the hippocampus with LPC. The effect of a 5-day stimulation was evaluated in postmortem brains. The work is interesting and shows relevant results, but in order to meet the standards for publication, authors need to address several issues. In the next lines, I present my comments, questions, and suggestions:

-          The introduction should be revised. Many of the cited works are very old, and a more updated view should be presented.

-          In the first paragraph, “Although the underlying mechanism of MS re-mains unclear, existing drugs used to treat MS target inflammation [2,3]. However, cur-rent efforts are focused on developing treatment approaches that utilize several combinations of compounds with different mechanisms of action with the aim of promoting re-myelination [4].” I consider that more details about the approved and the developing MS drugs are needed. Also, more comprehensive works have to be cited.

-          The last sentence of the first paragraph “Targeting these cells has the potential to treat progressive MS, which currently lacks effective therapies”. Author did not explain the MS forms, which I think they should do. In addition, they need to explain why they propose that targeting those cells would be relevant for progressive MS (and if so, why not for the relapsing forms).

-          “The pathogenesis of MS is characterized by a series of pathobiological events, ranging from microglia activation to demyelination and axonal degeneration [6,7]. Neurodegeneration in MS is commonly considered to be associated with the development of an inflammatory autoimmune response in the CNS [8]. Inflammation not only results in demyelination but also induces neuronal damage [9,10].” I think that this part should be presented before commenting about the drugs for MS (in the first paragraph).

-          There link between MMPs and the BDNF (which is mentioned afterwards) is not explained. Moreover, the manuscript does not investigate or propose any association with MMPs. There are many other components or processes implicated in MS pathology, so I do not understand why authors introduce only MMPs. Importantly, no reference is presented for the phrase “An increase in the activity of MMP-9 relative to TIMP-1 may promote formation of new MS lesions”.

-          In the introduction, another work with TUS in MS is mentioned: “Another study has demonstrated that TUS is capable of accelerating remyelination of MS lesions [26].” This work was carried out in the cuprizone model and with a different TUS protocol. I think that is a relevant work for the discussion section. It should be commented and compared to the results presented in the current manuscript.

-          The last sentence of the introduction is: “Our results suggest that TUS enhances remyelination in the hippocampus and corpus callosum of rats by upregulating BDNF levels and inhibiting neuroinflammatory activity.” But authors did not present any experiments or even mentioned again that they investigated the corpus callosum. Please, explain whether it was studied, and if so, what were the results in that area.

-          Regarding the animals used for the study, I believe that it is relevant to indicate and justify the age of the rats.

-          Similarly, TUS treatment protocol selection, including repetition frequency and duration should be explained. Authors mention previous works of the group, but in those different frequencies and durations were used. Besides, works from others, such as the one from reference n.26, also used a different approach. I think that authors should indicate whether other protocols were tested, and they should also discuss in the manuscript their opinion about the potential benefit of using other frequencies/durations and, in light of their results, what will be the next steps of their work.

-          The immunofluorescence protocol is vaguely described. More details about the procedure, the antibodies (references) and the used dilutions need to be included.

-          Authors indicated that “the remyelination index was calculated by dividing the value of the juxtaposed NF+ and MBP+ fraction by the value of the NF+ fraction.” If I understood this correctly, for instance, it would be about 1.2 divided by 9 for the sham group, so about 0.13, and this value is not the one presented in Figure2E.

-          Authors state that “Moreover, astrocytic activation was assessed by measuring the optical density of GFAP immunoactivity as previously described [27,34]”, but these the procedure to measure optical density is not described in those references. Please explain how OD is measured.

-          Then, the last sentence is: “The BDNF signals in the hippocampus of the treatment group were comparable to those signals in the sham group.” I consider that this should not be mentioned in the methods section, as it is a result. However, then we see in Figure 5 that BDNF signals are not the same in treated and sham groups, so this should be revised by authors.

-          Rat function was not monitored after LPC and during LIPUS, or these data is not presented. I would like the authors to comment on this point, as whether LPC treatment could alter behaviour and/or motor function of the animals.

-          Regarding the graphs, it would be much more informative to present the SD instead of the SEM, and to show all the data points instead of only bars (as authors did in Figure 4).

-          In the results section, authors wrote: “The evaluation of juxta-posed NF+ axons and MBP+ staining showed that the percentage of axons surrounded by visually healthy myelin sheaths significantly increased in LPC+LIPUS rats compared with LPC-only rats (0.88±0.03 versus 0.13±0.01, p<0.001; Figure 2D).” What do they mean by “visually healthy myelin”? And these data indicate that <1% of axons are surrounded by myelin? In addition, how do they explain that NF<10% and MBP about 30%? Does this mean that there is about 3 times more MBP than NF? Because this is not that we observe in Fig. 2A. Please, also revise the Y axes numbers in Fig. 2C, D and E, they seem to be offset.

-          “In the sham rats, saline injection into the hippocampus induced mild microglial activation (Figure 3F).” To demonstrate this, authors should compare microglia of the saline group to another group in which no injection was performed. “However, highly activated microglia were observed in the LPC-injected hippocampus compared with the saline-injected hippocampus (1 052.85±93.05 versus 67.79±3.75, p<0.001; Figure 3F). Microglial activation was manifested as an increase in the microglial density (Iba1+ cells).” In here only microglia density (number) was measured, so I believe that they could not say that microglia are “highly activated”.

-          Figure 3A. The name of the groups is not presented in the images.

-          Figure 3. For astrocytes, cell density, % of area and optical density are measured. In contrast, for microglia, only cell density is presented. Would it be interesting to show microglia area and optical density?

-          Regarding the different microglia phenotypes counts authors indicate that “In the demyelination lesion of the hippocampus, the number of microglia was counted in five randomly placed ROIs”. They were 5 ROIs per animal and the mean of each animal is shown in Fig. 4? I cannot see 5 different points in many groups of Fig. 4, particularly for the sham group.

-          If I understood correctly, Fig. 4G has the same information as Fig. 3F, total microglia. If this is the case, it should not be presented twice. If they are not the same, then I assume that some cells were not classified in the 5 microglia subgroups.

-          Authors mention astrocyte derived BDNF, but then they measure the general expression of BDNF, they cannot know which cells are producing it. This should be explained, or it could be misleading for readers. Moreover, the results about BDNF are presented in the subsection of oligodendrocytes, but no relation to these cells is mentioned.

-          In the discussion: “The current study results showed that LPC induction followed by LIPUS treatment reduced the extent of demyelination in the hippocampus…”. Previously in the manuscript authors were referring to an enhanced remyelination, which I think is correct. In contrast, I consider that they could not state that “LIPUS reduced the extent of demyelination”. LIPUS was applied 3 days after LPC treatment and tissue only analysed after one week. To test whether LIPUS reduces demyelination a different experimental protocol should be used. Similarly, I think that authors did not demonstrate that “LIPUS treatment inhibited the activation of astrocytes and microglia”. They could say that activation was reduced.

-          “The dystrophic and amoeboid phenotypes, which are typically associated with inflammatory responses [36],…” Reference 36 does not mention dystrophic microglia. Besides, “Compared to the LPS group, the hypertrophic and rod-shape phenotypes were significantly more abundant in the LPC+LIPUS group”. There is a small mistake and LPS must be substituted by LPC. More importantly, results are incorrectly presented: those two morphologies were LESS abundant in the LPC+LIPUS group. Finally, “Further studies are needed to investigate because there is currently no consensus-based agreement on definitions for the specific classes of microglia morphology.” This phrase does not make sense. Authors need to deeply revise the discussion about the microglia morphologies and comment the previous works that studied those morphologies.

-          Reference number 39 is a commentary about psychiatric disorders, I consider that is not relevant. Similarly, the sentence “Therefore, future works are necessary to investigate if TUS could be effective in inhibitory control for the treatment of psychiatric disorders [52,53]” does not add relevant information to the manuscript resented. I would like if authors could explain why they decided to include these.

-          Lastly, as commented for the results, also in the discussion authors wrote: “Therefore, we hypothesize that the LIPUS-stimulated release of astrocyte-derived BDNF promotes the upregulation of myelin protein levels that decrease after LPC injection.” This could not be hypothesized, because authors did not study BDNF produced by astrocytes.

Author Response

Comments and Suggestions for Authors

The work by Feng-Yi Yang and collaborators investigated the potential use of transcranial ultrasound stimulation (TUS) for multiple sclerosis. To this end, they used rats and induced demyelination in the hippocampus with LPC. The effect of a 5-day stimulation was evaluated in postmortem brains. The work is interesting and shows relevant results, but in order to meet the standards for publication, authors need to address several issues.

Explanation:

The manuscript has been revised as suggested. We sincerely appreciate the reviewers’ opinion to largely strengthen our manuscript. The made text changes with bold words are underlined in the amendment part. Moreover, all changes with red words are in the revised manuscript.

In the next lines, I present my comments, questions, and suggestions:

 -          The introduction should be revised. Many of the cited works are very old, and a more updated view should be presented.

-          In the first paragraph, “Although the underlying mechanism of MS re-mains unclear, existing drugs used to treat MS target inflammation [2,3]. However, cur-rent efforts are focused on developing treatment approaches that utilize several combinations of compounds with different mechanisms of action with the aim of promoting re-myelination [4].” I consider that more details about the approved and the developing MS drugs are needed. Also, more comprehensive works have to be cited.

Amendment:

The MS drugs have been added as suggested.

“Currently, more than a dozen drugs are approved for RRMS, and one agent for PPMS [11]. Glatiramer acetate is a well-studied synthetic copolymer that is approved for immune-based treatment of RRMS [12]. Ocrelizumab, a monoclonal antibody, was the first medication to be approved for PPMS [13].

-          The last sentence of the first paragraph “Targeting these cells has the potential to treat progressive MS, which currently lacks effective therapies”. Author did not explain the MS forms, which I think they should do. In addition, they need to explain why they propose that targeting those cells would be relevant for progressive MS (and if so, why not for the relapsing forms).

Amendment:

The descriptions have been added as suggested.

“MS can be classified according to its disease course into relapsing-remitting MS (RRMS), primary progressive MS (PPMS), and secondary progressive MS (SPMS) [2].

Targeting these cells has the potential to treat MS, which currently lacks effective therapies.

-          “The pathogenesis of MS is characterized by a series of pathobiological events, ranging from microglia activation to demyelination and axonal degeneration [6,7]. Neurodegeneration in MS is commonly considered to be associated with the development of an inflammatory autoimmune response in the CNS [8]. Inflammation not only results in demyelination but also induces neuronal damage [9,10].” I think that this part should be presented before commenting about the drugs for MS (in the first paragraph).

Amendment:

The part has been presented before commenting about the drugs for MS as suggested.

“The pathogenesis of MS is characterized by a series of pathobiological events, ranging from microglia activation to demyelination and axonal degeneration [2, 3]. Neurodegeneration in MS is commonly considered to be associated with the development of an inflammatory autoimmune response in the CNS [4]. Inflammation not only results in demyelination but also induces neuronal damage [5, 6].”

-          There link between MMPs and the BDNF (which is mentioned afterwards) is not explained. Moreover, the manuscript does not investigate or propose any association with MMPs. There are many other components or processes implicated in MS pathology, so I do not understand why authors introduce only MMPs. Importantly, no reference is presented for the phrase “An increase in the activity of MMP-9 relative to TIMP-1 may promote formation of new MS lesions”.

Amendment:

In fact, one reviewer suggested that we should briefly introduce the MMPs in introduction while we submitted last journal. But, we also think that MMPs are not investigated in this study. Therefore, the following MMPs descriptions have been deleted as suggested.

“Increasing evidences suggest that matrix metalloproteinases (MMPs) are involved in various processes of MS pathogenesis [15, 16]. Moreover, the activity of MMPs may be controlled by specific tissue inhibitors of those enzymes (TIMPs) [17]. An increase in the activity of MMP-9 relative to TIMP-1 may promote formation of new MS lesions.”

-          In the introduction, another work with TUS in MS is mentioned: “Another study has demonstrated that TUS is capable of accelerating remyelination of MS lesions [26].” This work was carried out in the cuprizone model and with a different TUS protocol. I think that is a relevant work for the discussion section. It should be commented and compared to the results presented in the current manuscript.

Amendment:

The previous results have been compared in the discussion section.

“Compared to previous study [27], we demonstrated that LIPUS not only increase the myelin in the demyelination lesion, but also enhance oligodendrocyte maturation.”

-          The last sentence of the introduction is: “Our results suggest that TUS enhances remyelination in the hippocampus and corpus callosum of rats by upregulating BDNF levels and inhibiting neuroinflammatory activity.” But authors did not present any experiments or even mentioned again that they investigated the corpus callosum. Please, explain whether it was studied, and if so, what were the results in that area.

Amendment:

Here, the corpus callosum was not studied. The mistake has been corrected.

“Our results suggest that TUS enhances remyelination in the hippocampus of rats by upregulating BDNF levels and inhibiting neuroinflammatory activity.

-          Regarding the animals used for the study, I believe that it is relevant to indicate and justify the age of the rats.

Amendment:

The age of rats has been indicated in the method.

Sprague Dawley rats weighing from 280 to 320 g (age: 12 weeks)”

-          Similarly, TUS treatment protocol selection, including repetition frequency and duration should be explained. Authors mention previous works of the group, but in those different frequencies and durations were used. Besides, works from others, such as the one from reference n.26, also used a different approach. I think that authors should indicate whether other protocols were tested, and they should also discuss in the manuscript their opinion about the potential benefit of using other frequencies/durations and, in light of their results, what will be the next steps of their work.

Amendment:

The description has been added in the discussion.

“Although increasingly being investigated, TUS is still in its early phases. Further investigations of optimal ultrasound parameters are needed.

-          The immunofluorescence protocol is vaguely described. More details about the procedure, the antibodies (references) and the used dilutions need to be included.

Amendment:

The more details have been added as suggested.

MBP (1:400; Abcam, Cambridge, MA, USA), neurofilament (NF; 1:50; Abcam, Cambridge, MA, USA), BDNF (1:200; Abcam, Cambridge, MA, USA), and markers of microglia (ionized calcium-binding adaptor molecule 1 [Iba1]; 1:200; GeneTex, Inc., California, USA), astrocytes (glial fibrillary acidic protein [GFAP]; 1:200; GeneTex, Inc., California, USA), OPCs (Olig2; 1:100; Abcam, Cambridge, MA, USA), and mature oligodendrocytes (CC-1; 1:200; Sigma-Aldrich, St. Louis, Mo., USA) [28, 35].

-          Authors indicated that “the remyelination index was calculated by dividing the value of the juxtaposed NF+ and MBP+ fraction by the value of the NF+ fraction.” If I understood this correctly, for instance, it would be about 1.2 divided by 9 for the sham group, so about 0.13, and this value is not the one presented in Figure2E.

Amendment:

The % of the sham has been revised.

The Figure 2E has been revised and the ratio of value has been presented in Figure 2E as suggested.

-          Authors state that “Moreover, astrocytic activation was assessed by measuring the optical density of GFAP immunoactivity as previously described [27,34]”, but these the procedure to measure optical density is not described in those references. Please explain how OD is measured.

Amendment:

The procedure to measure optical density was described in reference [36].

“Moreover, astrocytic activation was assessed by measuring the optical density of GFAP immunoactivity as previously described [28, 36].”

-          Then, the last sentence is: “The BDNF signals in the hippocampus of the treatment group were comparable to those signals in the sham group.” I consider that this should not be mentioned in the methods section, as it is a result. However, then we see in Figure 5 that BDNF signals are not the same in treated and sham groups, so this should be revised by authors.

Amendment:

This sentence has been deleted as suggested.

-          Rat function was not monitored after LPC and during LIPUS, or these data is not presented. I would like the authors to comment on this point, as whether LPC treatment could alter behaviour and/or motor function of the animals.

Explanation:

The goal of this study was to investigate the possible effects in the demyelination lesion after LIPUS treatment by

histological analysis. Behavioral or motor function will be investigated in the next step.

-          Regarding the graphs, it would be much more informative to present the SD instead of the SEM, and to show all the data points instead of only bars (as authors did in Figure 4).

Amendment:

The SEM has been changed to SD and show all data points instead of only bars as suggested.

“Figure 2, Figure 3, and Figure 5”

-          In the results section, authors wrote: “The evaluation of juxta-posed NF+ axons and MBP+ staining showed that the percentage of axons surrounded by visually healthy myelin sheaths significantly increased in LPC+LIPUS rats compared with LPC-only rats (0.88±0.03 versus 0.13±0.01, p<0.001; Figure 2D).” What do they mean by “visually healthy myelin”? And these data indicate that <1% of axons are surrounded by myelin? In addition, how do they explain that NF<10% and MBP about 30%? Does this mean that there is about 3 times more MBP than NF? Because this is not that we observe in Fig. 2A. Please, also revise the Y axes numbers in Fig. 2C, D and E, they seem to be offset.

Explanation:

Yes. These data indicate that <1% of axons are surrounded by myelin. This means that there is about 3 times more MBP than NF.

Amendment:

Visually healthy myelin has been revised to healthy myelin.

The Fig. 2A has been improved. The Y axes numbers in Fig. 2C, D and E have been revised.

-          “In the sham rats, saline injection into the hippocampus induced mild microglial activation (Figure 3F).” To demonstrate this, authors should compare microglia of the saline group to another group in which no injection was performed. “However, highly activated microglia were observed in the LPC-injected hippocampus compared with the saline-injected hippocampus (1 052.85±93.05 versus 67.79±3.75, p<0.001; Figure 3F). Microglial activation was manifested as an increase in the microglial density (Iba1+ cells).” In here only microglia density (number) was measured, so I believe that they could not say that microglia are “highly activated”.

Amendment:

The descriptions have been revised as suggested.

“In the sham rats, saline injection into the hippocampus showed mild microglial activation (Figure 3F).”

“However, a significant increase of activated microglia were observed in the LPC-injected hippocampus compared with the saline-injected hippocampus”

-          Figure 3A. The name of the groups is not presented in the images.

Amendment:

The names of the groups in figure 3A have been added as suggested.

“ Sham, LPC, LPC+LIPUS ”

-          Figure 3. For astrocytes, cell density, % of area and optical density are measured. In contrast, for microglia, only cell density is presented. Would it be interesting to show microglia area and optical density?

Explanation:

Yes. We found that the trend of microglia and optical density is similar, but no significant. Therefore, we investigate the microglial phenotypes in Fig.4.

-          Regarding the different microglia phenotypes counts authors indicate that “In the demyelination lesion of the hippocampus, the number of microglia was counted in five randomly placed ROIs”. They were 5 ROIs per animal and the mean of each animal is shown in Fig. 4? I cannot see 5 different points in many groups of Fig. 4, particularly for the sham group.

Explanation:

Yes. They were 5 ROIs per animal and the mean of each animal is shown in Fig. 4.

Amendment:

The graph has been revised in Fig. 4. There are five animals for each group. Thus, there are 5 different points in each group in Fig. 4.

-          If I understood correctly, Fig. 4G has the same information as Fig. 3F, total microglia. If this is the case, it should not be presented twice. If they are not the same, then I assume that some cells were not classified in the 5 microglia subgroups.

Amendment:

Yes. Fig. 4G has the same information as Fig. 3F, total microglia. The Fig. 4G has been deleted as suggested.

-          Authors mention astrocyte derived BDNF, but then they measure the general expression of BDNF, they cannot know which cells are producing it. This should be explained, or it could be misleading for readers. Moreover, the results about BDNF are presented in the subsection of oligodendrocytes, but no relation to these cells is mentioned.

Explanation:

Yes, we discuss the relation in the discussion section.

“BDNF increases myelinogenesis and axonal regeneration by promoting the differentiation of OPCs [43]. The upregulation of MBP expression is thought to be activated by BDNF signaling to enhance the differentiation of OPCs. A significant increase in oligodendrocyte proliferation is observed following BDNF administration in cases of spinal cord injury [44]. We observed an increase in BDNF (Figure 5E) and MBP (Figure 2C) expression following the LIPUS treatment. In a previous study, this effect was found to be dependent upon the release of BDNF by GFAP+ astrocytes, suggesting that astrocyte activation may promote recovery from demyelination by providing neurotrophic support [16].”

-          In the discussion: “The current study results showed that LPC induction followed by LIPUS treatment reduced the extent of demyelination in the hippocampus…”. Previously in the manuscript authors were referring to an enhanced remyelination, which I think is correct. In contrast, I consider that they could not state that “LIPUS reduced the extent of demyelination”. LIPUS was applied 3 days after LPC treatment and tissue only analysed after one week. To test whether LIPUS reduces demyelination a different experimental protocol should be used. Similarly, I think that authors did not demonstrate that “LIPUS treatment inhibited the activation of astrocytes and microglia”. They could say that activation was reduced.

Amendment:

The descriptions have been revised as suggested.

“LIPUS treatment enhanced the extent of remyelination in the hippocampus”

“LIPUS treatment reduced the activation of astrocytes and microglia…”

-          “The dystrophic and amoeboid phenotypes, which are typically associated with inflammatory responses [36],…” Reference 36 does not mention dystrophic microglia. Besides, “Compared to the LPS group, the hypertrophic and rod-shape phenotypes were significantly more abundant in the LPC+LIPUS group”. There is a small mistake and LPS must be substituted by LPC. More importantly, results are incorrectly presented: those two morphologies were LESS abundant in the LPC+LIPUS group. Finally, “Further studies are needed to investigate because there is currently no consensus-based agreement on definitions for the specific classes of microglia morphology.” This phrase does not make sense. Authors need to deeply revise the discussion about the microglia morphologies and comment the previous works that studied those morphologies.

Amendment:

The reference [39] has been added for dystrophic microglia.

“The dystrophic and amoeboid phenotypes, which are typically associated with inflammatory responses [38, 39],”

The mistakes have been corrected.

“Compared to the LPC group, the hypertrophic and rod-shape phenotypes were significantly decreased in the LPC+LIPUS group.”

The phrase ‘Further studies…’ has been deleted as suggested.

-          Reference number 39 is a commentary about psychiatric disorders, I consider that is not relevant. Similarly, the sentence “Therefore, future works are necessary to investigate if TUS could be effective in inhibitory control for the treatment of psychiatric disorders [52,53]” does not add relevant information to the manuscript resented. I would like if authors could explain why they decided to include these.

Amendment:

In fact, one reviewer suggested that we should add these references and the inhibitory control in discussion while we submitted last journal. But, we also think that they are not investigated in this study. Therefore, the reference 42 about psychiatric disorders and the following descriptions have been deleted as suggested.

“Therefore, future works are necessary to investigate if TUS could be effective in inhibitory control for the treatment of psychiatric disorders [55, 56].”

-          Lastly, as commented for the results, also in the discussion authors wrote: “Therefore, we hypothesize that the LIPUS-stimulated release of astrocyte-derived BDNF promotes the upregulation of myelin protein levels that decrease after LPC injection.” This could not be hypothesized, because authors did not study BDNF produced by astrocytes.

Amendment:

The descriptions have been deleted as suggested.

“Therefore, we hypothesize that the LIPUS-stimulated release of astrocyte-derived BDNF promotes the upregulation of myelin protein levels that decrease after LPC injection.”

Round 2

Reviewer 2 Report

I would like to thank the authors for their efforts in improving the manuscript. They notably improved the document and I believe that it could be published. However, there are still some issues that need to be addressed. Here are my new comments:

1. Authors have improved the presentation of MS therapies, but: “Currently, more than a dozen drugs are approved for RRMS, and one agent for PPMS [11]. Glatiramer acetate is a well-studied synthetic copolymer that is approved for immune-based treatment of RRMS [12]. Ocrelizumab, a monoclonal antibody, was the first medication to be approved for PPMS [13].” I can understand that they mention ocrelizumab, but I do not see the reason for mentioning glatiramer acetate, there are many other immune-based treatments for MS.

2. This sentence has been modified, but it is not correct: “Targeting these cells has the potential to treat MS, which currently lacks effective therapies.” I believe that authors cannot state that MS lacks effective treatments, patients benefit from the treatments available. Please revise the sentence and include an appropriate explanation about the effectiveness of MS therapies.

3. Going back to my previous comments and the response of authors:

In the introduction, another work with TUS in MS is mentioned: “Another study has demonstrated that TUS is capable of accelerating remyelination of MS lesions [26].” This work was carried out in the cuprizone model and with a different TUS protocol. I think that is a relevant work for the discussion section. It should be commented and compared to the results presented in the current manuscript.

Amendment:

The previous results have been compared in the discussion section.

“Compared to previous study [27], we demonstrated that LIPUS not only increase the myelin in the demyelination lesion, but also enhance oligodendrocyte maturation.”

I think that authors did not completely address my suggestion. I would like them to mention that the previous study was carried out with a different animal model and protocol, and that they briefly discuss the different results and implications.

4. Again, my previous comment was not addressed:

Similarly, TUS treatment protocol selection, including repetition frequency and duration should be explained. Authors mention previous works of the group, but in those different frequencies and durations were used. Besides, works from others, such as the one from reference n.26, also used a different approach. I think that authors should indicate whether other protocols were tested, and they should also discuss in the manuscript their opinion about the potential benefit of using other frequencies/durations and, in light of their results, what will be the next steps of their work.

Amendment:

The description has been added in the discussion.

“Although increasingly being investigated, TUS is still in its early phases. Further investigations of optimal ultrasound parameters are needed.”

I would like to ask the authors to explicitly indicate whether they tested other protocols, why/how they chose the used protocol, their opinion about the potential benefit of using other frequencies/durations and, what will be the next steps of their work.

5. Regarding optical density, the references in the new version of the manuscript – n28 and 36 – still do not explain how OD was measured.

6. In response to my previous comment, authors state that “The goal of this study was to investigate the possible effects in the demyelination lesion after LIPUS treatment by histological analysis. Behavioral or motor function will be investigated in the next step.” I think that this should be mentioned in the manuscript. It is an important point for the readers.

7. Regarding Figure 2:

“Explanation:

Yes. These data indicate that <1% of axons are surrounded by myelin. This means that there is about 3 times more MBP than NF.

Amendment:

Visually healthy myelin has been revised to healthy myelin.

The Fig. 2A has been improved. The Y axes numbers in Fig. 2C, D and E have been revised.”

I would like to thank the authors for improving the figure. However, I cannot see in the images that there is about 3 times more MBP than NF. Maybe the thresholds for the area quantification of NF were more restrictive?

Besides, I consider that authors cannot talk about “healthy myelin”. They do not know whether it is healthy, they are only measuring MBP staining.

8. My previous comments about BDNF have not been completely addressed. Authors mention and cite both in the results and the discussion previous works about astrocyte derived BDNF. However, they measure the general expression of BDNF, they cannot know which cells are producing it. This should be indicated in the text, or it could be misleading for readers.

Finally, I would like to suggest that for future works authors use the “Track Changes” function in their word processor, so that reviewers can clearly see what parts of the text have been, deleted, added or modified.

Author Response

Comments and Suggestions for Authors

I would like to thank the authors for their efforts in improving the manuscript. They notably improved the document and I believe that it could be published. However, there are still some issues that need to be addressed.

Explanation:

The manuscript has been revised as suggested. We sincerely appreciate the reviewers’ opinion to largely strengthen our manuscript. The made text changes with bold words are underlined in the amendment part. Moreover, all changes with red words are in the revised manuscript.

Here are my new comments:

  1. Authors have improved the presentation of MS therapies, but: “Currently, more than a dozen drugs are approved for RRMS, and one agent for PPMS [11]. Glatiramer acetate is a well-studied synthetic copolymer that is approved for immune-based treatment of RRMS [12]. Ocrelizumab, a monoclonal antibody, was the first medication to be approved for PPMS [13].” I can understand that they mention ocrelizumab, but I do not see the reason for mentioning glatiramer acetate, there are many other immune-based treatments for MS.

Amendment:

The description of glatiramer acetate has been deleted as suggested.

  1. This sentence has been modified, but it is not correct: “Targeting these cells has the potential to treat MS, which currently lacks effective therapies.” I believe that authors cannot state that MS lacks effective treatments, patients benefit from the treatments available. Please revise the sentence and include an appropriate explanation about the effectiveness of MS therapies.

Amendment:

The MS drugs have been added as suggested.

“Targeting these cells, along with currently available disease-modifying therapies, has the potential to develop a therapeutic strategy to treat progressive forms of MS, which currently lacks effective therapies.”

  1. Going back to my previous comments and the response of authors:

In the introduction, another work with TUS in MS is mentioned: “Another study has demonstrated that TUS is capable of accelerating remyelination of MS lesions [26].” This work was carried out in the cuprizone model and with a different TUS protocol. I think that is a relevant work for the discussion section. It should be commented and compared to the results presented in the current manuscript.

Amendment:

The previous results have been compared in the discussion section.

“Compared to previous study [27], we demonstrated that LIPUS not only increase the myelin in the demyelination lesion, but also enhance oligodendrocyte maturation.”

I think that authors did not completely address my suggestion. I would like them to mention that the previous study was carried out with a different animal model and protocol, and that they briefly discuss the different results and implications.

Amendment:

The descriptions have been added as suggested.

“Compared to previous study [26], we demonstrated that LIPUS not only increase the myelin in the demyelination lesion, but also enhance oligodendrocyte maturation. The previous work was carried out in the cuprizone model and with a different ultrasound protocol. Consequently, it might be that at different parameters and animal models, different mechanisms exist for coupling acoustic wave into brain activity.

  1. Again, my previous comment was not addressed:

Similarly, TUS treatment protocol selection, including repetition frequency and duration should be explained. Authors mention previous works of the group, but in those different frequencies and durations were used. Besides, works from others, such as the one from reference n.26, also used a different approach. I think that authors should indicate whether other protocols were tested, and they should also discuss in the manuscript their opinion about the potential benefit of using other frequencies/durations and, in light of their results, what will be the next steps of their work.

Amendment:

The description has been added in the discussion.

“Although increasingly being investigated, TUS is still in its early phases. Further investigations of optimal ultrasound parameters are needed.”

I would like to ask the authors to explicitly indicate whether they tested other protocols, why/how they chose the used protocol, their opinion about the potential benefit of using other frequencies/durations and, what will be the next steps of their work.

Amendment:

The discussion has been added as suggested.

“Although increasingly being investigated, TUS is still in its early phases. An ultrasound stimulus is defined by several parameters: operating frequency, intensity, duration, and pulse repetition frequency. Each of these parameters may have different effect on experimental outcome. In this study, one set of parameters was selected based on our previous works. Further investigations of optimal ultrasound parameters are needed.”

  1. Regarding optical density, the references in the new version of the manuscript – n28 and 36 – still do not explain how OD was measured.

Amendment:

The references have been revised in the method as suggested.

“…measuring the optical density of GFAP immunoactivity as previously described [35, 36].”

  1. In response to my previous comment, authors state that “The goal of this study was to investigate the possible effects in the demyelination lesion after LIPUS treatment by histological analysis. Behavioral or motor function will be investigated in the next step.” I think that this should be mentioned in the manuscript. It is an important point for the readers.

Amendment:

The descriptions have been mentioned in the discussion as suggested.

Besides, the goal of this study was to investigate the possible effects in the demyelination lesion after LIPUS treatment by histological analysis. Behavioral or motor function will be investigated in the next step.

  1. Regarding Figure 2:

“Explanation:

Yes. These data indicate that <1% of axons are surrounded by myelin. This means that there is about 3 times more MBP than NF.

Amendment:

Visually healthy myelin has been revised to healthy myelin.

The Fig. 2A has been improved. The Y axes numbers in Fig. 2C, D and E have been revised.”

I would like to thank the authors for improving the figure. However, I cannot see in the images that there is about 3 times more MBP than NF. Maybe the thresholds for the area quantification of NF were more restrictive?

Besides, I consider that authors cannot talk about “healthy myelin”. They do not know whether it is healthy, they are only measuring MBP staining.

Explanation:

The images that there is about 2-3 times more MBP than NF.

Amendment:

The healthy myelin has been revised to myelin as suggested.

  1. My previous comments about BDNF have not been completely addressed. Authors mention and cite both in the results and the discussion previous works about astrocyte derived BDNF. However, they measure the general expression of BDNF, they cannot know which cells are producing it. This should be indicated in the text, or it could be misleading for readers.

Finally, I would like to suggest that for future works authors use the “Track Changes” function in their word processor, so that reviewers can clearly see what parts of the text have been, deleted, added or modified.

Amendment:

In the results: The sentence below has been deleted to avoid misleading for readers.

“LIPUS treatment has been reported to increase the production and release of BDNF from astrocytes [19].”

In the discussion: The sentence below has been deleted to avoid misleading for readers.

“In a previous study, this effect was found to be dependent upon the release of BDNF by GFAP+ astrocytes, suggesting that astrocyte activation may promote recovery from demyelination by providing neurotrophic support [15].”

In addition, another sentence has been added for clearer discussion.

A bunch of evidence indicates that BDNF can promote myelin regeneration in different animal models of demyelination [45, 46].